# Effects of Carbon Dioxide Therapy on Skin Wound Healing

**DOI:** 10.3390/biomedicines13010228

**Published:** 2025-01-18

**Authors:** José Prazeres, Ana Lima, Gesiane Ribeiro

**Affiliations:** 1I-MVET Research in Veterinary Medicine, Faculty of Veterinary Medicine, Lusófona University—Lisbon University Centre, 1749-024 Lisbon, Portugal; vetstoestevao@gmail.com (J.P.); p6256@ulusofona.pt (A.L.); 2Veterinary and Animal Research Centre (CECAV), Faculty of Veterinary Medicine, Lusófona University—Lisbon University Centre, Campo Grande 376, 1749-024 Lisbon, Portugal

**Keywords:** carbon dioxide, CO_2_, carboxytherapy, healing, skin, wound

## Abstract

Promoting rapid healing is a concern in skin wound treatment, as the increased pain and the loss of functional ability when wounds become chronic create a complex problem to manage. This scoping review aimed to explore the literature and synthesize existing knowledge on the therapeutic use of CO_2_ in treating cutaneous wounds. The literature was selected using previously defined inclusion and exclusion criteria, and 22 articles were selected for data extraction. The most researched type of injury was chronic wounds located on the extremities of the limbs. Carboxytherapy was performed in five different ways: subcutaneous, intradermal, or intralesional injections; in hot water baths with temperatures ranging from 30 to 42 °C; transcutaneous application; intra-abdominal insufflation; and a paste for transcutaneous local application. The main effects of CO_2_ therapy described were as follows: improved blood flow and local oxygenation, reduction of the inflammatory process, increased collagen production, and improved clinical aspects of wounds, with faster healing. Carboxytherapy can be considered a good alternative for treating skin wounds, although further studies should be pursued to elucidate its molecular mechanisms and enhance its efficacy.

## 1. Introduction

Chronic or difficult-to-heal wounds significantly impact health systems, and the primary wound-related healthcare resources include hospital outpatient visits, practice nurse visits, hospitalizations, healthcare assistants home visits, laboratory tests, medications, and dressing materials [1,2]. The quality of life of affected individuals is also impacted in terms of function, mobility, daily activities, and mental health [3]. In veterinary medicine, they are also a frequent cause of euthanasia [4].

The wound healing process is complex, involving four overlapping phases: hemostasis, inflammation, proliferation, and maturation. Chronic wounds are those in which the healing process does not occur normally, either in time or in the sequence of phases. They are more common on the extremities but are also associated with other co-morbidities [5].

The therapeutic application of gases has been used in wound treatment, including oxygen (O_2_), hyperbaric oxygen (HBO), nitric oxide (NO), hydrogen sulfide (H_2_S), ozone (O_3_), carbon monoxide (CO), and carbon dioxide (CO_2_) [6]. The latter has been used in different administration forms, such as bathing in water enriched with CO_2_, transdermal administration, and subcutaneous injection [7].

Currently, CO_2_ therapy is attracting attention in various fields, such as health [8], beauty and well-being [9], and sports [10], due to the therapeutic effects attributed to an increase in the partial pressure of oxygen in the local area, known as the Bohr effect, where elevated CO_2_ levels lower the blood’s affinity for oxygen, facilitating its release into tissues. However, the detailed mechanism of these therapeutic effects is not yet completely understood [11]. Exploring CO_2_’s effect on critical processes, such as endothelial cell function, fibroblast proliferation, and immune cell activity, could provide mechanistic insights that not only enhance our understanding but also optimize its application in clinical wound healing.

A scoping review is a specific type of review that uses a systematic and well-defined method to identify and synthesize existing literature on a given topic [12]. This study aimed to explore the literature and provide a scoping review of current knowledge resulting from research and clinical application of CO_2_ in the treatment of skin wounds. We hypothesize that CO_2_ therapy may help treat cutaneous wounds, especially chronic wounds associated with circulatory disorders. From a scoping review, it is possible to understand what exists in the literature on this topic and identify knowledge gaps to direct future research.

## 2. Materials and Methods

This scoping review was prepared according to the framework described by Arksey and O’Malley (2005) [13], following the Preferred Reporting Items for Systematic Reviews and Meta-Analyses for Scoping Reviews (PRISMA-ScR) [14], and the methodological updates for scoping reviews [15].

### 2.1. Identifying the Research Question

“What do we know about the therapeutic use of carbon dioxide in skin wound management?” was the research question proposed for this scoping review.

### 2.2. Identifying Relevant Studies

The search strategy used was as follows: skin wound AND (healing OR treatment OR management) AND (CO_2_ OR carbon dioxide) NOT laser. Search filter was used for languages (English, French, Spanish, and Portuguese). The search was performed on July 2024, and the databases used were Pubmed, Web of Science, and Scopus.

### 2.3. Study Selection

All articles found in the three databases were imported into Ryyan software (https://rayyan.qcri.org/welcome, accessed on 5 July 2024). Mak and Thomas (2022) [16] recommended this software as a good tool to identify duplicates and help in the first screening level, allowing researchers to analyze articles individually and blindly.

Studies describing the therapeutic use of carbon dioxide in treating skin wounds were included. Articles selected could have addressed protocols for all types of wounds, and any form of carboxytherapy application, in all species, in vivo or in vitro. The types of articles included were as follows: experimental, case reports and case series, and retrospective.

The research addressing aesthetic purposes such as rejuvenation or wrinkle treatment were excluded. CO_2_ laser treatments were also excluded.

The articles underwent two stages of screening and one of data extraction. The first screening was performed by analyzing the titles and abstracts, and the second by reading the articles in full. Two researchers performed screening and data extraction independently and resolved conflicts by consensus.

### 2.4. Charting the Data

Two researchers independently reviewed the full articles and extracted the following data:-Authors and date;-Manuscript language;-Study location;-Study design;-Participant count;-Species;-Type of wound;-Wound location;-Treatments (protocol details);-Outcomes measured;-Results (intervention effects).

### 2.5. Collating, Summarizing, and Reporting Data

The extracted data were presented descriptively and in tables. When appropriate, figures were also used to summarize the data.

## 3. Results

The results of the database search and the flow of articles through the selection process for inclusion in the scoping review are presented in Figure 1. Four hundred ninety-nine papers were found (Pubmed: 201, Scopus: 183, and Web of Science: 115) and 148 were duplicated. After the stage 1 screening, 26 articles were identified as potentially relevant, and 9 of these records were excluded at level 2 because they did not exactly address the use of CO_2_ for treating skin wounds. Five studies not found in the initial search but which met the eligibility criteria were included.

The 22 articles eligible for the scoping review were published between 2000–2024, and the number of articles per year was as follows: 2000 (1), 2003 (1), 2004 (1), 2008 (1), 2010 (1), 2011 (1), 2012 (1), 2013 (2), 2015 (2), 2018 (2), 2020 (4), 2021 (1), 2022 (1), 2023 (2), and 2024 (1).

The greatest number of publications was from Brazil (*n* = 6), followed by Japan (*n* = 5), Germany (*n* = 4), Egypt, Hungary, Italy, Jordan, Slovenia, Spain, and Turkey (*n* = 1) (Figure 2). Most of the articles were in English (*n* = 21), only one in Portuguese, and none in French or Spanish. Although the articles are written in English, the lack of contributions from areas such as North America or the United Kingdom is curious.

Regarding the study design, the largest number of manuscripts were experimental studies (14/22), followed by case reports (5/22). Two articles were case series and only one was a retrospective study (Figure 3). This result is expected, since new therapies require experimental studies to better understand and standardize protocols before clinical trials can be initiated. Experimental studies add value to the strength of evidence, but clinical reports also provide constraints and clues for further investigations.

The species studied in the included manuscripts were humans (14/22), rats (6/22), mice (1/22), and rabbits (1/22). The case reports, case series, and retrospective study analyzed injuries in humans. Of the 14 experimental studies, 7 analyzed injuries in rats, 6 in humans, 1 in mice, and 1 in rabbits. The average number of individuals participating in experimental studies was 45, with a minimum number of 9 and a maximum of 96 participants.

The anatomical location and type of injury varied between manuscripts. In experimental studies conducted on humans, the most researched type of injury was chronic wounds located at the extremities of the limbs. In experimental research with rats, the type of skin injury was incisional or made with a Punch and the most common region was the back. Among case reports and case series, ulcers on limbs also prevailed, followed by heat press injury. The retrospective study compared chronic wounds and acute surgical wounds, in different areas.

Carboxytherapy was performed in five different ways: subcutaneous, intradermal, or intralesional injections; in hot water baths with temperatures ranging from 30 to 42 °C; transcutaneous application; intra-abdominal insufflation; and a paste for transcutaneous local application. The most frequent form of administration was injectable, described in 8/22 studies, almost all experimental. In case reports, the most common carboxytherapy was through bathing (5/22) and transcutaneous administration (5/22). The effect of CO_2_ pneumoperitoneum on skin wound healing was evaluated in three experimental studies and the use of CO_2_ paste was described in one study (Figure 4.)

The “outcomes measured” indicate which tools were used in the studies to assess the effectiveness of the treatment performed. Among the manuscripts included in this review, clinical evaluation was the main tool used. As a clinical evaluation, we included all macroscopic evaluations of the wounds, such as the size, color, presence of abscess and foreign body, evaluation of the appearance of the wound bed, sensation, and patient satisfaction. Of the 22 articles analyzed, 9 used only clinical evaluation to validate the effectiveness of the treatment described. These articles included all case reports, case series, and retrospective study, and only one experimental study. In addition to clinical evaluation, all other experimental studies used additionally tools such as histological and molecular exams, blood flow by a Doppler flowmeter, and the tensile strength of the healing wound.

Regarding the results obtained in the studies included in this review, Table 1 summarizes the data extracted. In general, the manuscripts presented good results with the use of CO_2_ for the treatment of skin wounds, regardless of the route of administration. Studies that evaluated the effects of the pneumoperitoneum reported that the use of CO_2_ does not interfere with the healing process of the abdominal wall or distant injuries (such as the back). One of these studies compared abdominal insufflation with CO_2_, helium, and air, finding beneficial effects on healing with helium pneumoperitoneum.

## 4. Discussion

Carbon dioxide therapy refers to CO_2_ administration for therapeutic purposes, and, to produce the desired effects, an adequate amount of CO_2_ needs to be delivered to local tissues without difficulty and invasion. Three methods of CO_2_ administration that were able to accomplish this are as follows: bathing in CO_2_-enriched water, direct subcutaneous CO_2_ injection, and the transcutaneous administration of CO_2_ [7].

According to Frangež et al. (2021) [39], the role of natural water rich in CO_2_ is known for its positive effects on wound healing and it is used more frequently than gaseous CO_2_. However, the form of medical application has been modified in recent decades, as the transcutaneous application of gaseous CO_2_ avoids hydration of the wound and the inhalation of CO_2_ evaporated from water. All case reports included in this review showed positive effects with the use of carboxytherapy, both in the form of bathing [24,25,26] and transcutaneous application [27,37,38]. The evaluation of the treatment was carried out clinically and the main effects found were as follows: the removal of necrotic tissue, an improvement in granulation, a reduction in secretion, and bad odor.

The majority of experimental studies included in this review used injection as a form of CO_2_ administration. Subcutaneous CO_2_ injection, although it can provide pure CO_2_ in local tissues, is an invasive method, that involves the risk of infection, and it is difficult to use over a large area of the body [7]. Additionally, some patients found the treatments quite painful [36]. In addition to the improvement in clinical aspects described in case reports, experimental studies observed significantly faster healing in wounds treated with CO_2_ compared to control groups. Faster healing was observed with hot spring water rich in carbonate ion [28], with transcutaneous application of CO_2_ gas [29], and also with a CO_2_ paste covering the wound [19].

The improvement in clinical aspects and faster healing may be related to the better blood perfusion in wounds treated with CO_2_. According to Finzgar et al. (2015) [40], the inefficient healing of chronic wounds is the result of poor blood perfusion in the wound and surrounding tissues and artificially applied carbon dioxide has the potential to improve tissue perfusion and oxygenation, therefore being useful for the healing of chronic wounds. In fact, Shalan et al. (2015) [35] reported improved blood flow in the feet of patients after CO_2_ therapy, and Liang et al. (2015) [28] showed histologically increased vessel density in wounds treated with thermal water rich in carbonate ion compared to control groups. The significant increase in tissue oxygenation values was confirmed by the better evolution of the lesions, both in the improvement in healing and in the reduction in the injured area in the group treated with CO2 [20].

The Bohr effect is characterized by the stimulation of dissociation between oxygen and hemoglobin (Hb), causing the release of oxygen into the blood when there is an increase in the concentration of carbon dioxide; that is, when CO_2_ levels become high, the blood’s affinity for oxygen decreases, facilitating its release into the tissues. Near-infrared spectroscopy was used to confirm that the transcutaneous application of CO_2_ actually causes the dissociation of O_2_ from oxy-Hb, which is a characteristic phenomenon of the Bohr effect. The experimental results showed scientific evidence that the transcutaneous application of CO_2_ can cause an “artificial Bohr effect”. This artificial Bohr effect could be a potential new therapy for disorders in which a high amount of O_2_ in local tissues is required for treatment, as well as in peripheral vascular disorder [7].

In 2020, Oda et al. [41] investigated whether CO_2_ treatment would promote fracture repair in cases with type I Diabetes mellitus. This study showed that the gene expression levels of vascular endothelial growth factor (VEGF) in newly generated callus tissue were significantly higher in the CO_2_ group at all time points. Through fluorescent immunostaining with isolectin B4, the researchers also observed that angiogenesis around the fracture was stimulated in the CO_2_ group. Therefore, the results showed that CO_2_ therapy reverses the reduced levels of VEGF gene expression in the fracture region and improves angiogenesis, contributing to fracture repair. Kuroiwa et al. (2019) [11] also reported that the expression of the VEGF gene in the CO_2_ group was significantly higher than in the control group.

VEGF plays a critical role in wound healing by promoting angiogenesis and enhancing oxygen and nutrient delivery to hypoxic tissues. Various therapeutic approaches, such as the administration of recombinant growth factors (e.g., VEGF-A or PDGF-BB) [42,43], the use of biomaterials [44], and mesenchymal stem cell (MSC)-based therapies [45], have been developed to upregulate VEGF expression and improve tissue regeneration. Compared to these strategies, CO_2_ treatment offers a singular method to stimulate VEGF’s expression and enhances angiogenesis without the need for exogenous growth factors or complex interventions.

Collagen gene expression levels have also been reported to be significantly higher in the CO_2_ group than in the control group [41]. Oliveira et al. (2020) [32] observed an increase in collagen and elastic fibers in the group treated with a single application of carboxytherapy with an infusion rate of 100 mL/min and 0.6 mL/kg of weight in an area of 25 cm^2^ in the abdominal region in humans. The use of CO_2_ injection around grafts also increased the amount of collagen in 2 cm grafts in rabbits [23].

Some studies included in this review reported a decreased inflammatory process in wounds treated with CO_2_ in rats [21], and human [31]. In 2021, Sayama et al. [46] clarified the anti-inflammatory mechanism of CO_2_ using a UVB-induced inflammation model. The researchers observed that CO_2_ can decrease the production of IL-6 and TNFα in human keratinocytes and the 3D epidermis, thereby decreasing the formation of UVB-induced erythema in human skin. TNFα and IL-6 are cytokines that play essential roles in skin inflammation. Therefore, CO_2_ can reduce UV-induced inflammation by decreasing the production of these inflammatory cytokines.

Although further clinical studies are needed to quantify the effects of TNF-α and IL-6 suppression on specific clinical metrics, such as pain scores, wound size reduction, and edema resolution, the overall literature suggests that CO_2_ therapy can benefit clinical outcomes such as wound pain and edema. Firstly, studies show that CO_2_ therapy suppresses TNF-α and IL-6 production, thereby reducing the inflammatory burden in both animal models and human subjects [46]. Usually, elevated levels of these cytokines are associated with pain, tissue damage, and edema, as they enhance vascular permeability, recruit inflammatory cells, and amplify nociceptive signaling pathways [47,48]. The suppression of TNF-α and IL-6 in CO_2_ treatments correlates with clinical observations of decreased erythema, reduced swelling, and improved granulation tissue formation. Moreover, the attenuation of inflammation likely contributes to lower nociceptive signaling, as suggested by the documented improvement in patient-reported discomfort and overall wound condition in case reports [47]. Edema may also be alleviated through CO_2_ therapy due to its effects on tissue oxygenation, blood flow, and vascular leakage reduction, as observed in studies such as Shalan et al. (2015) [35], all of which may directly counteract edema formation and facilitate its resolution.

CO_2_ can permeate through the stratum corneum and promote mild extracellular acidification by reacting with H_2_O in the interstitial fluid and producing H^+^. Considering this, it was investigated whether the anti-inflammatory effects of CO_2_ could be associated with this change in extracellular pH. In vitro studies using human keratinocytes revealed that the CO_2_-induced suppression of pro-inflammatory cytokines TNF-a and IL-6 was pH-dependent. These findings suggest that extracellular pH is crucial in modulating skin inflammation [46].

Changes in the intracellular and extracellular pH can exert different physiological effects. In the skin, it is known that the pH of the stratum corneum plays an important role in several pathological conditions. It is generally maintained in an acidic range of 4.1–5.8; however, there is an increase in inflammatory skin diseases [49]. Previous research has demonstrated that changes in pH lead to the dysregulation of several skin functions, including antimicrobial defense mechanisms, barrier action, and inflammatory responses. Thus, the topical application of CO_2_ may enhance the stratum corneum’s barrier and antimicrobial properties while attenuating the epidermis’s excessive inflammatory reactions through CO_2_-induced acidification [46].

Another study [16] demonstrates that CO₂ paste promotes wound healing by modulating the hypoxic environment, reducing inflammation, and accelerating angiogenesis. Specifically, it enhanced the upregulation of growth factors such as VEGF, TGF-β, and platelet-derived growth factor (PDGF), which, in turn, stimulate angiogenesis and fibroblast proliferation. CO_2_ also decreased the levels of interleukins (IL-1β and IL-6), which are critical for regulating the inflammatory phase and promoting cellular recruitment. The signaling cascade involved the activation of nitric oxide synthase (NOS) and the production of nitric oxide (NO), along with downregulation of Hypoxia-Inducible Factor 1-alpha (HIF-1α), indicating an improved oxygenation status in the wound area. These pathways converge to accelerate granulation tissue formation, epithelialization, and wound closure.

Concerning TGF-β and VEGF, the literature shows they play complementary roles in distinct but interdependent phases of the process. On the one hand, TGF-β regulates the inflammatory and proliferative phases of wound healing, promoting fibroblast migration, extracellular matrix deposition, and myofibroblast differentiation necessary for wound contraction and closure [50]. On the other hand, VEGF mainly facilitates angiogenesis, increasing vascular density and oxygen supply to the wound site, which are critical for the reparative processes and epithelialization [51]. Although its contributions are well-established at the molecular level, linking them to clinical outcomes such as wound closure can be challenging as clinical outcomes are influenced by multiple factors (such as patient comorbidities, and wound type and location), but, overall, CO_2_ therapy has been shown to have the potential to modulate TGF-β and VEGF in a manner that aligns with physiological healing processes.

Overall, while the results of existing studies on CO_2_ in wound healing are promising, the current body of research remains limited. Future work should focus on elucidating the precise molecular mechanisms, particularly the interactions between CO_2_, inflammatory cytokines, growth factors, and hypoxia-inducible factors. Important clinical information such as the dosage, duration of exposure, and administration methods should also be further addressed to enhance efficacy and safety.

## 5. Conclusions

Although carboxytherapy is widely used for different purposes, the existing literature on the use of CO_2_ as a specific treatment for skin wounds is not that extensive. The vast majority of articles report the beneficial effects of the use of CO_2_ on skin healing, in all forms of use. The main effects of carboxytherapy described were the following: an improvement in blood flow and local oxygenation, a reduction in the inflammatory process, increased collagen production, and an improvement in the clinical aspects of wounds, with faster healing. Given this, the use of carboxytherapy in treating skin wounds seems very promising, and it should be studied in more depth to better understand the mechanisms of action and the effects obtained with the treatment, as well as to define the best protocol for clinical use.

## Figures and Tables

**Figure 1 biomedicines-13-00228-f001:**
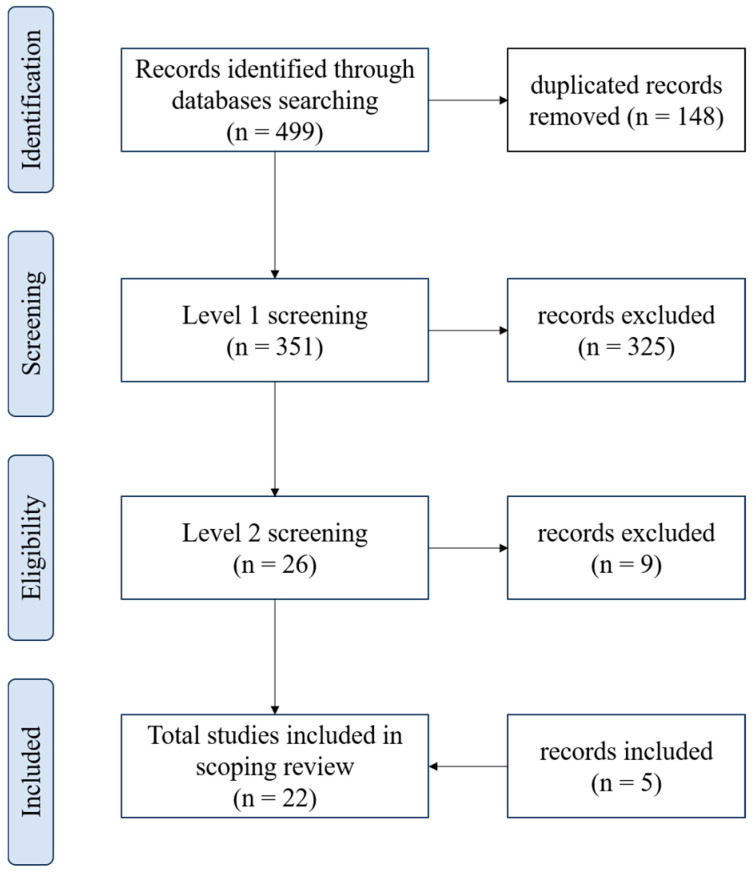
Flowchart showing the process of selecting articles included in the scoping review, according to PRISMA.

**Figure 2 biomedicines-13-00228-f002:**
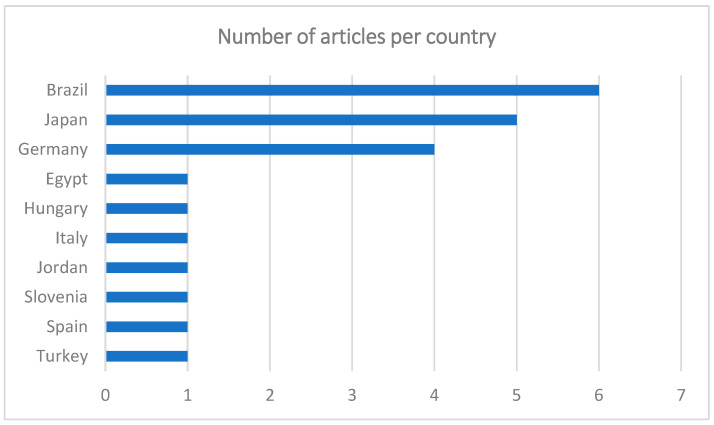
Graph showing the number of articles per country.

**Figure 3 biomedicines-13-00228-f003:**
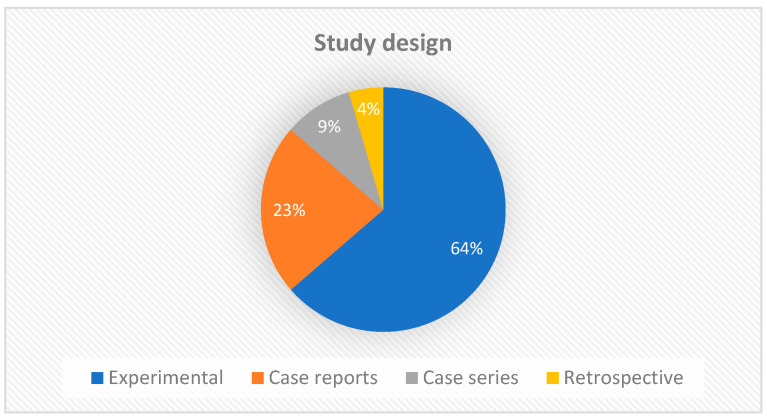
Graph showing the types of studies included in the scoping review.

**Figure 4 biomedicines-13-00228-f004:**
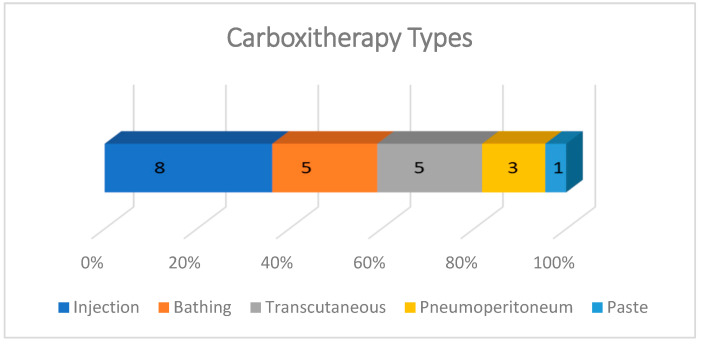
Graph showing the types of carboxytherapy described in studies included in the scoping review. The most frequent forms of application were injection, bathing, and transcutaneous. The intraperitoneal route was used only in experiments with rats involving laparotomy, and the CO_2_-rich paste was tested in only one experimental study with rats.

**Table 1 biomedicines-13-00228-t001:** Summary of data extracted from the 22 articles included in the scoping review about the effects of carboxytherapy on skin wound healing.

**Authors and Date**	**Study Design**	**Species**	** *n* **	**Treatments**	**Results**
Abramo & Teixeira, 2011 [17]	Experimental	Humans	10	CO_2_ infusion—controlled was applied 0.5 cm from the wound edge. Punctures were carried out at 2.5 cm distance from each other along the border of the wound with the needle pointed toward the granulation tissue. Biopsies were taken from the granulation tissue before and after CO_2_ application.	Noticeable increase in capillaries, macrophages, and fibroblasts were found in the granulation tissue after CO_2_ infusion—controlled. The diameter of pre-capillary arterioles increased 3.4 times after CO_2_ infusion—controlled.
Agalar et al., 2000 [18]	Experimental	Mice	72	Control (dorsal skin incision) x Laparotomy (dorsal skin incision + laparotomy) x CO_2_ insufflation (dorsal skin incision + CO_2_ intra-abdominal insufflation).	There was no significant difference between the tensile strengths of the incised skin of control, laparotomy, and CO_2_ insufflation groups throughout the observation period.
Amano-Iga et al., 2021 [19]	Experimental	Rats	48	CO_2_ paste covering the wound for 10 min every day after surgery x untreated (control).	Rats in the CO_2_ group showed accelerated wound healing compared to those in the control group.
Brandi et al., 2010 [20]	Experimental	Humans	70	Routine methods of treatment + subcutaneous administration of CO_2_ with 30 G/13 mm needles twice a week for six weeks x routine methods alone.	Significant increase in tissue oxygenation values, which was confirmed by greater progress of the lesions in terms of both healing and reduction of the injured area in the group treated with CO_2_.
Brochado et al., 2018 [21]	Experimental	Rats	96	Daily cleaning with 0.9% saline solution x 1% silver sulfadiazine x subcutaneous application of 0.3 mL of carbon dioxide.	Carboxytherapy decreased the inflammatory process, and improved the restructuring of the basal membrane through greater synthesis and better organization of collagen.
Csonka et al., 2020 [22]	Case Report	Humans	1	Surgical debridement and jet irrigation. Synthetic skin substitute. Antibiotic, low-molecular-weight heparin, and antihypertensive treatments. Vacuum-assisted closure therapy. Bilayer matrix wound dressing. Polarized light therapy. D’Oxyva deoxyhemoglobin vasodilator. CO_2_ administration transcutaneously.	Following three and a half months of inpatient care, the affected area of the skin was macroscopically healed, and the patient was discharged home. The case emphasizes the importance of soft-tissue care and a multidisciplinary approach in diabetic trauma patients with poor compliance.
Durães et al., 2013 [23]	Experimental	Rabbits	20	CO_2_ injection surrounding grafts x saline solution injection (control).	No significant difference in graft survival rate was found. The use of CO_2_ increases the amount of collagen in 2 cm graft.
Hihara et al., 2022 [24]	Case Report	Humans	1	Daily foot bath in carbonated water (AS care; Asahi Kasei Medical Co., Ltd., Tokyo, Japan) at 37 °C with a concentration of 1000–3000 ppm for 15 min, followed by applying povidone–iodine sugar ointment.	After 6 months of biweekly wound examinations, wound closure, and bone and joint remodeling were observed, and the therapy was concluded.
Hihara et al., 2023 [25]	Case Report	Humans	1	From the day after the injury, a 15 min hand bath in highly concentrated carbon dioxide bathing at 37 °C (AS Care^®^; Asahi Kasei Medical Co., Ltd., Tokyo, Japan) was performed daily.	4 weeks after the injury, the palmar necrotic tissue had been completed eliminated and sufficient granulation tissue had grown.
Hihara et al., 2024 [26]	Case series	Humans	3	Highly concentrated carbon dioxide bathing (AS Care®; Asahi Kasei Medical Co., Ltd., Tokyo, Japan) at 37 °C for 15 min once daily.	Highly concentrated CO_2_ bathing was sufficient to prevent wound infections. On average, 13 weeks of treatment led to significantly better wound bed preparation.
Hohaus et al., 2003 [27]	Case Report	Humans	1	Topical CO_2_ gas was applied using the Carboflow™ device (Carboflow, Gernsbach, Germany) for 30 min every day. The area was covered by a plastic bag to ensure the persistence of the gas above the wounds.	The wounds healed with a good clinical and functional result.
Liang et al., 2015 [28]	Experimental	Rats	44	Hot spring water (42 °C) rich in carbonate ion x unbathed (control) x hot-water (42 °C) control.	Hot spring water (42 °C) rich in carbonate ion led to an enhanced healing speed compared to both the unbathed and hot-water (42 °C) control groups. Histologically, this showed increased vessel density and reduced inflammatory cells in the granulation tissue of the wound area.
Macura et al., 2020 [29]	Experimental	Humans	43	Standard treatment + Transcutaneous application using Peripheral Vascular Rehabilitation system (PVR system; Derma Art, Brežice, Slovenia) of therapeutic concentration 99.9% gaseous CO_2_ for 45 min x the same treatment with air.	Significantly faster healing in the study group, mean wound surface, and volume in the study group were reduced significantly compared with a small reduction in the control group.
Morais et al., 2012 [30]	Experimental	Rats	80	CO_2_ pneumoperitoneum for 30 min before laparotomy x CO_2_ pneumoperitoneum for 30 min after abdominal closure x CO_2_ pneumoperitoneum for 30 min before laparotomy and 30 min after abdominal closure x control group without CO_2_ pneumoperitoneum.	There were no differences in histopathology and in tensile strength values between experimental and control groups. CO_2_ pneumoperitoneum did not interfere with abdominal wall wound healing.
Nassar et al., 2020 [31]	Experimental	Humans	40	PRP injected intralesionally and subcutaneously with 1 cm between injection points x CO_2_ intradermal injection with 2 mL of gas per injection spot, with flow rate 40 to 100 cc/min. Treatment every 4 weeks for four sessions.	Clinical improvement, patients’ satisfaction, and significant expression of MMP-1 with carboxytherapy group.
Oliveira et al., 2020 [32]	Experimental	Humans	9	A single application of carboxytherapy with an infusion rate of 100 mL/min and 0.6 mL/kg weight over an area of 25 cm^2^ x control.	An increase in the collagen and elastic fibers was observed in the treated group.
Penhavel et al., 2013 [33]	Experimental	Rats	16	Subcutaneous injections of carbon dioxide on the day of operation and at three, six, and nine days postoperatively x no postoperative wound treatment (control).	There was no difference between groups in the wound area and histopathological findings at 14 days postoperatively.
Rosch et al., 2007 [34]	Experimental	Rats	58	CO_2_ pneumoperitoneum before and after laparotomy with a total duration of 30 min x helium pneumoperitoneum x control (abdominal cavity was exposed to room air for 30 min).	Infiltration of macrophages and expression of MMP-13 were greatest in helium pneumoperitoneum group. Collagen I/III ratio was significantly increased in the helium group. Results suggest beneficial effects on systemic wound healing for helium pneumoperitoneum as compared to CO_2_.
Shalan et al., 2015 [35]	Experimental	Humans	22	The patients immersed their feet in a disposable bag containing carbon dioxide (1000 ppm) dissolved in water at 37 °C for 30 min each session. These sessions were repeated daily for 15 days. The wounds were evaluated before and after CO_2_ therapy.	Improvement in blood flow to the affected foot as well as improvement in the sensation and color of the ulcerative area.
Waked et al., 2023 [36]	Case series	Humans	5	CO_2_ heated to 30 °C injection, with a flow rate of 30–60 mL/min, every day or every other day for 1 or 2 weeks.	Significant clinical improvement. In 3 of the 5, there was a complete skin recovery at 3 months. Patients found the treatments quite painful.
Wollina et al., 2004 [37]	Retrospective	Humans	86	Transdermal CO_2_ application was performed with a Carboflow^®^ device once daily for 30 to 60 min until improvement in the wound was evidenced (either complete healing or significant decrease in inflammation, discharge, pain, and malodor). In chronic wounds, the treatment was usually performed 6 to 14 days, and, in acute wounds, 5 to 12 days.	Improvement in granulation and reduction in discharge and malodor within 1 week of treatment in both chronic and acute wounds. Only 9 patients, all diabetics, needed an additional systemic antibiosis. The treatment was well-tolerated. No adverse effects have been noted.
Wollina et al., 2018 [38]	Case Report	Humans	1	Deep ulcer shaving in combination with sandwich meshed graft transplantation. Anti-septic wound cover, multiple necrosectomies, and daily transcutaneous application of CO_2_ for 30 min.	The patient had a 100% graft taken with rapid reduction in severe wound pain.

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
