# Peer review of "Effects of Carbon Dioxide Therapy on Skin Wound Healing"

_biomedicines, 2025, doi:10.3390/biomedicines13010228_

Round 1
Reviewer 1 Report
Comments and Suggestions for Authors
I had the privilege to go through the manuscript titled “Effects of CO2 on Skin Wound Healing” which is a decent effort to provide a review of the current knowledge regarding carboxytherapy of chronic wounds. In my view it is a manuscript that worths to be considered for publication although some amendments may be considered.
In brief
Line 2 it may be considered to remove the CO2 form the title either substituting it with “carbon dioxide therapy” or “carboxytherapy” since the scope of the manuscript is to describe the effects of the therapeutic application of CO2 and not the CO2 biology in wound healing in general.
Lines 8 – 24 A careful rewrite of the abstract may be considered to improve precision. A moderate English language review may also be considered.
Line 37 Consider changing “The application of gases” by “The therapeutic application of gases” or “Gas therapy”
Lines 39 – 40 The sentence “The latter has been widely used since ancient times due to its therapeutic effects and the use of thermal waters” seems to lack justification. In my view the therapeutic application of gases is a method strongly connected to the technology of pure gas preservation that is a matter of the last two centuries. How was that performed in antiquity? In any case, my view could not be true, but a literature citation should be cited here for that. Note that thermal spring water was implemented as therapeutic media not only for wounds from the antiquity, but its therapeutic effects are not attributed solely to CO2 content. In toto such therapeutic practice, in my view, is largely different from carboxytherapy.
Lines 44 - 47 Citations are needed after the words “sports” “beauty” , “health” and “tissues”. Moreover, these lines seem to provide one key element of the review namely the “why” this therapeutic intervention may work but the Bohr effect is not well. Lines 213 - 221 may be transposed here.
Lines 53 – 55 The scope of the study must be described better, possibly describing also the view of the authors about the importance of the method examined (why is that important for our patients and why it is worth to look towards this intervention)
Line 62 Consider “therapeutic use”
Line 69 Why was not the term Carboxytherapy (AND wound healing) included?
Line 181 – 183 The authors mention “Three methods of CO2 administration that were able to accomplish this are: bathing in CO2-enriched water, direct subcutaneous CO2 injection, and transcutaneous administration of CO2” Besides this and a small subsequent part I was not able to identify any other part of the review were tit is described what is ”carboxytherapy” how is it performed, which between these tree methods is better? How the dose is established? Ecc. In my view that should be a basic part of the review. From a clinical point of view no intervention procedures are reviewed.
Lines 281 – 286 The conclusion part although valid is somehow narrow, not providing insights about the views of the authors on the findings of their literature search. They mention solely that 1. Literature is limited 2. That there is beneficial effect from this method 3. That the mechanisms are those mentioned. But the research presented is not only about that. It is also about if it worths to continue research on this method. It is about the possible benefots of our patients. It is also about the quality of the literature on this matter (Were the articles reviewed of good quality?)
Lastly, this review seems to be limited by several factors, such as the small size of literature, and the quality of papers reviewed (large part of them are case studies). As most studies have limitations, any limitations should be clearly stated by the authors in a separate paragraph in the discussion part.
Overall, in my view the manuscript is a bit confusing from an organizational point of view. In my view. Animal studies should be presented in different paragraphs than the human ones. The possible mechanisms should be summarized in different subparagraph all together. And there should be subparagraphs about the exact methods and the therapeutic outcomes (stating the results of comparative studies were available). The view of the authors and the future perspectives are also a largely missing part.
But I am convinced that this manuscript worths to be considered for publication in its revised form since it constituted a good addition to international literature.
Author Response
Dear Reviewer,
Thank you very much for taking the time to review this manuscript. Please find the detailed responses below and the corresponding revisions/corrections highlighted in the re-submitted file.
Line 2: it may be considered to remove the CO2 form the title either substituting it with “carbon dioxide therapy” or “carboxytherapy” since the scope of the manuscript is to describe the effects of the therapeutic application of CO2 and not the CO2 biology in wound healing in general.
Response: We agree with the suggestion. The correction has been made.
Lines 8 - 24: A careful rewrite of the abstract may be considered to improve precision. A moderate English language review may also be considered.
Response: We agree with the suggestion. The abstract has been rewritten.
Line 37: Consider changing “The application of gases” by “The therapeutic application of gases” or “Gas therapy”.
Response: We agree with the suggestion. The correction has been made.
Lines 39 – 40: The sentence “The latter has been widely used since ancient times due to its therapeutic effects and the use of thermal waters” seems to lack justification. In my view the therapeutic application of gases is a method strongly connected to the technology of pure gas preservation that is a matter of the last two centuries. How was that performed in antiquity? In any case, my view could not be true, but a literature citation should be cited here for that. Note that thermal spring water was implemented as therapeutic media not only for wounds from the antiquity, but its therapeutic effects are not attributed solely to CO2 content. In toto such therapeutic practice, in my view, is largely different from carboxytherapy.
Response: We agree with the suggestion. The sentence has been rewritten.
Lines 44 – 47: Citations are needed after the words “sports” “beauty” , “health” and “tissues”. Moreover, these lines seem to provide one key element of the review namely the “why” this therapeutic intervention may work but the Bohr effect is not well. Lines 213 - 221 may be transposed here.
Response: We partially agree with the suggestions. The requested citations have been included. Lines 213 to 221 were not transposed to the introduction because this paragraph discusses the data extracted in this scoping review and should not be placed before the presentation of the results.
Lines 53 – 55: The scope of the study must be described better, possibly describing also the view of the authors about the importance of the method examined (why is that important for our patients and why it is worth to look towards this intervention)
Response: We agree with the suggestion. A paragraph has been added.
Line 62: Consider “therapeutic use”
Response: We agree with the suggestion. The term “therapeutic use” was added.
Line 69: Why was not the term Carboxytherapy (AND wound healing) included?
Response: Several combinations of terms were tested before the final search with the guidance of a librarian. The phrase “skin wound AND (healing OR treatment OR management) AND (CO2 OR carbon dioxide) NOT laser” was chosen because it was the combination of terms that resulted in the largest number of articles.
Line 181 – 183: The authors mention “Three methods of CO2 administration that were able to accomplish this are: bathing in CO2-enriched water, direct subcutaneous CO2 injection, and transcutaneous administration of CO2” Besides this and a small subsequent part I was not able to identify any other part of the review were tit is described what is ”carboxytherapy” how is it performed, which between these tree methods is better? How the dose is established? Ecc. In my view that should be a basic part of the review. From a clinical point of view no intervention procedures are reviewed.
Response: We understand that the questions raised are important and could be researched in a systematic review. However, a scoping review is not intended to provide these answers.
“A scoping review is a relatively new approach to evidence synthesis and differs from systematic reviews in its purpose and aims. The purpose of a scoping review is to provide an overview of the available research evidence without producing a summary answer to a guide clinical decision-making.” https://med.cornell.libguides.com/systematicreviews/scopingreviews
Lines 281 – 286: The conclusion part although valid is somehow narrow, not providing insights about the views of the authors on the findings of their literature search. They mention solely that 1. Literature is limited 2. That there is beneficial effect from this method 3. That the mechanisms are those mentioned. But the research presented is not only about that. It is also about if it worths to continue research on this method. It is about the possible benefits of our patients. It is also about the quality of the literature on this matter (Were the articles reviewed of good quality?)
Response: We have followed the guidelines for preparing a scoping review and in this type of review, authors should not critically analyze the articles. “Scoping reviews aim to provide a descriptive overview of the reviewed material without critically appraising individual studies or synthesizing evidence from different studies (no risk of bias or meta-analysis/statistical pooling is performed).” https://med.cornell.libguides.com/systematicreviews/scopingreviews
Thank you for the suggestions.
Best regards,
Gesiane Ribeiro et al.
Reviewer 2 Report
Comments and Suggestions for Authors
I would want to thank the writers for giving me the chance to check over this scoping review paper. This is a thorough and orderly attempt to investigate how carbon dioxide treatment—carboxytherapy—may affect wound healing. Given growing interest in creative and non-invasive techniques for treating complicated and chronic wounds, the subject is contemporary and pertinent. The writers should be praised for their careful reading of the literature and integration of results between several study types and species. With line numbers for clarity, I have thorough remarks below split between big and small changes.
Major Comments
Although the search technique is detailed in line 56–70, it is not obvious how the search phrases were generated or verified. Ask whether expert consultations or pilot searches guided the search keywords.
Line 91-92 (Data Extraction) Although the variables taken from trials are thorough, there is no reference to how qualitative data—that is, patient experiences or side effects—were handled. Consider including a quick discussion on qualitative data management.
Line 104–113 (PRISMA Flowchart): Although the text's numerical values—that is, the count of eliminated duplicates—would improve clarity and match the visual data, the flowchart is presented nicely.
While the distribution of research types is obvious, an explanation of why experimental studies dominate the review and the constraints of depending on case reports would add depth. Line 123–127 (research Design Breakdown).
Line 144–148: Carboxytherapy Types:
Although Figure 4's bar graph is useful, placing numerical values on the bars would make it even more instructive. Note why particular techniques—such as injection—are chosen for use in experimental investigations.
Line 178- 201: Discussion—Application Techniques:
The debate stresses the variety of approaches but does not examine which, in therapeutic environments, are most useful or efficient. Think about compiling trends from several research.
Line 202-207 (Improvement Mechanisms): Although a major explanation is discussed—improved perfusion—no other cause is investigated in circumstances when perfusion had no bearing. Think about closing these divisions.
Line 229-238: VEGGF Expression
Although the debate about VEGF overexpression is interesting, it does not provide background on how these results fit other wound-healing treatments. Comparative study would improve the part.
Line 264–266: Although TGF-β and VEGF are discussed, their exact contribution to clinical outcomes—that is, wound closure rates—is not entirely clear. Comment on their translational relevance.
Line 276–278: Although general future objectives are discussed, particular research subjects or approaches are not suggested. More detail would make this part of the workable.
Minor Comments
Line 28–31 (Introduction): As this gives the relevance of the research a practical aspect, consider momentarily discussing how chronic wounds affect healthcare expenses.
Line 62–65: Here the reference to Lusófona University is superfluous and might divert the reader. Emphasize the research question itself.
Line 117–120: To set the geographical distribution, one may perhaps mention the lack of contributions from areas such as North America or the UK.
Line 149–150: The figure caption may have a remark outlining the reasons behind the less often investigated pneumoperitoneum and pasting techniques in carboxytherapy graph.
Line 152–156: To simplify the description, think about grouping the evaluation techniques—that is, molecular, histological, and clinical ones.
Case Reports: The particular techniques used in transcutaneous treatments are covered; but, a quick discussion of limitations—that is, time or financial restrictions—would be beneficial.
Line 213–217 (Bohr Effect): Although the Bohr effect is clearly described, a condensed explanation for those not familiar with biology would help the material to be more easily available.
Line 242–247: Although the study of TNF-α and IL-6 suppression is important, it would be helpful to link these results to certain clinical outcomes (e.g., lowered wound pain or edema).
Line 254–259 ( Skin pH): Although the function of pH is clarified, no useful advice is given on how CO2 treatment should be maximized to sufficiently change pH. Think about adding such observations.
Line 282–284: The conclusion nicely summarizes results but should finish with more of an emphasis on the clinical consequences and scalability of CO2 treatment.
The English writing in this paper is clean and professional, with a suitable academic tone. However, there are several small points where readability and accuracy might be enhanced. These contain some repetition, unnecessarily complicated language, and minor grammatical faults. I urge that you proofread carefully to resolve these flaws and enhance the general flow and clarity.
Author Response
Dear Reviewer,
Thank you very much for taking the time to review this manuscript. Please find the detailed responses below and the corresponding revisions/corrections highlighted in the re-submitted file.
Line 56-70: Although the search technique is detailed in line 56–70, it is not obvious how the search phrases were generated or verified. Ask whether expert consultations or pilot searches guided the search keywords.
Response: Several combinations of terms were tested before the final search with the guidance of a librarian. The phrase “skin wound AND (healing OR treatment OR management) AND (CO2 OR carbon dioxide) NOT laser” was chosen because it was the combination of terms that resulted in the most significant number of articles.
Line 91-92 (Data Extraction): Although the variables taken from trials are thorough, there is no reference to how qualitative data—that is, patient experiences or side effects—were handled. Consider including a quick discussion on qualitative data management.
Response: Our intention with this study was to map the existing literature and provide an overview of this field. Scoping reviews are descriptive rather than analytical.
https://med.cornell.libguides.com/systematicreviews/scopingreviews
When patient experiences or side effects were reported in the articles, they were presented as treatment outcomes (Table 1. Results column of Shalan et al., 2015 ; Waked et al., 2023 ; Wollina et al., 2004 ; Wollina et al., 2018).
Line 104–113 (PRISMA Flowchart): Although the text's numerical values—that is, the count of eliminated duplicates—would improve clarity and match the visual data, the flowchart is presented nicely.
Response: The number of duplicates eliminated (n=148) is presented in the text (line 105) and the flowchart (Figure 1).
Line 123-127: While the distribution of research types is obvious, an explanation of why experimental studies dominates the review and the constraints of depending on case reports would add depth. Line 123–127 (research Design Breakdown).
Response: We agree with the suggestion. A paragraph has been added.
Line 144–148: Carboxytherapy Types: Although Figure 4's bar graph is useful, placing numerical values on the bars would make it even more instructive. Note why particular techniques—such as injection—are chosen for use in experimental investigations.
Response: We agree with the suggestion. Numerical values ​​were placed on the bars of the graph.
Line 178-201: Discussion—Application Techniques: The debate stresses the variety of approaches but does not examine which, in therapeutic environments, are most useful or efficient. Think about compiling trends from several research.
Response: Our intention with this study was to map the existing literature and provide an overview of this field. Scoping reviews are not analytical.
“The purpose of a scoping review is to provide an overview of the available research evidence without producing a summary answer to a guide clinical decision-making.”
https://med.cornell.libguides.com/systematicreviews/scopingreviews
Line 202-207 (Improvement Mechanisms): Although a major explanation is discussed—improved perfusion—no other cause is investigated in circumstances when perfusion had no bearing. Think about closing these divisions.
Response: In addition to the improvement in blood flow, the improvement in tissue oxygenation promoted by the Bohr effect was also discussed, as well as the increased gene expression levels of VEGF and collagen, the decreased inflammatory process by suppressed TNFα and IL-6 production, and the effects of intracellular and extracellular pH changes.
Line 229-238: VEGF Expression. Although the debate about VEGF overexpression is interesting, it does not provide background on how these results fit other wound-healing treatments. Comparative study would improve the part.
Response: We agree that providing a comparative context for the VEGF overexpression observed in CO2 treatments would strengthen this section. We have included these considerations in the revised manuscript to address the broader context of VEGF modulation in wound healing and how CO2 therapy compares to other interventions, and we hope it meets your requirements.
Line 264–266: Although TGF-β and VEGF are discussed, their exact contribution to clinical outcomes—that is, wound closure rates—is not entirely clear. Comment on their translational relevance.
Response: Thank you for your observation. We have added a new paragraph concerning the translational relevance of TGF-β and VEGF to wound closure rates.
Line 276–278: Although general future objectives are discussed, particular research subjects or approaches are not suggested. More detail would make this part of the workable.
Response: Future work should focus on molecular mechanisms, particularly the interaction between CO2, inflammatory cytokines, growth factors, hypoxia-inducible factors, and clinical research such as dosage, exposure duration, and delivery methods, as already mentioned in our study.
Minor Comments
Line 28–31 (Introduction): As this gives the relevance of the research a practical aspect, consider momentarily discussing how chronic wounds affect healthcare expenses.
Response: We agree with the suggestion. The paragraph has been rewritten in more detail.
Line 62–65: Here the reference to Lusófona University is superfluous and might divert the reader. Emphasize the research question itself.
Response: We agree with the suggestion. This information has been removed.
Line 117–120: To set the geographical distribution, one may perhaps mention the lack of contributions from areas such as North America or the UK.
Response: We agree with the suggestion. This observation was mentioned.
Line 149–150: The figure caption may have a remark outlining the reasons behind the less often investigated pneumoperitoneum and pasting techniques in carboxytherapy graph.
Response: We agree with the suggestion. A paragraph has been added.
Line 152–156: To simplify the description, think about grouping the evaluation techniques—that is, molecular, histological, and clinical ones.
Response: We agree with the suggestion. The description has been simplified.
Case Reports: The particular techniques used in transcutaneous treatments are covered; but, a quick discussion of limitations—that is, time or financial restrictions—would be beneficial.
Response: Articles that used transcutaneous CO2 application did not report any restrictions related to this form of use.
Line 213–217 (Bohr Effect): Although the Bohr effect is clearly described, a condensed explanation for those not familiar with biology would help the material to be more easily available.
Response: Thank you for your observation. We agree with the suggestion, and a condensed explanation has been added.
Line 242–247: Although the study of TNF-α and IL-6 suppression is important, it would be helpful to link these results to certain clinical outcomes (e.g., lowered wound pain or edema).
Response: Thank you for your insightful comment. We added more text that links these cytokines and clinical outcomes, such as wound pain and edema.
Line 254–259 ( Skin pH): Although the function of pH is clarified, no useful advice is given on how CO2 treatment should be maximized to sufficiently change pH. Think about adding such observations.
Response: We understand the search for helpful advice for clinical applications. However, a scoping review aims to provide an overview of the available research evidence without producing a summary answer to guide clinical decision-making.
Line 282–284: The conclusion nicely summarizes results but should finish with more of an emphasis on the clinical consequences and scalability of CO2 treatment.
Response: We agree with the suggestion. A paragraph has been added to improve the conclusion.
Comments on the Quality of English Language
The English writing in this paper is clean and professional, with a suitable academic tone. However, there are several small points where readability and accuracy might be enhanced. These contain some repetition, unnecessarily complicated language, and minor grammatical faults. I urge that you proofread carefully to resolve these flaws and enhance the general flow and clarity.
Response: Thank you for the suggestions. The article has been completely revised to improve the English.
Best regards,
Gesiane Ribeiro et al.
Round 2
Reviewer 1 Report
Comments and Suggestions for Authors
Dear authors
Thank you for considering my suggentions
Reviewer 2 Report
Comments and Suggestions for Authors
Dear Authors,
Thanks for addressing the comments and the opportunity to review your manuscript.
I have no more comments.
Best of luck